# *Pseudomonas putida* Biofilm Depends on the vWFa-Domain of LapA in Peptides-Containing Growth Medium

**DOI:** 10.3390/ijms23115898

**Published:** 2022-05-24

**Authors:** Marge Puhm, Johanna Hendrikson, Maia Kivisaar, Riho Teras

**Affiliations:** Institute of Molecular and Cell Biology, University of Tartu, 51010 Tartu, Estonia; marge.puhm@gmail.com (M.P.); johanna.hendrikson@ut.ee (J.H.); maia.kivisaar@ut.ee (M.K.)

**Keywords:** *Pseudomonas putida*, LapA, Fis, peptides, vWFa, biofilm

## Abstract

The biofilm of *Pseudomonas putida* is complexly regulated by several intercellular and extracellular factors. The cell surface adhesin LapA of this bacterium is a central factor for the biofilm and, consequently, the regulation of *lapA* expression, for example, by Fis. It has been recently shown that peptides in growth media enhance the formation of *P. putida* biofilm, but not as a source of carbon and nitrogen. Moreover, the peptide-dependent biofilm appeared especially clearly in the *fis*-overexpression strain, which also has increased LapA. Therefore, we investigate here whether there is a relationship between LapA and peptide-dependent biofilm. The *P. putida* strains with inducible *lapA* expression and LapA without the vWFa domain, which is described as a domain similar to von Willebrand factor domain A, were constructed. Thereafter, the biofilm of these strains was assessed in growth media containing extracellular peptides in the shape of tryptone and without it. We show that the vWFa domain in LapA is necessary for biofilm enhancement by the extracellular peptides in the growth medium. The importance of vWFa in LapA was particularly evident for the *fis*-overexpression strain F15. The absence of the vWFa domain diminished the positive effect of Fis on the F15 biofilm.

## 1. Introduction

The two adhesins of *Pseudomonas putida*, LapA and LapF, are responsible for attachment, initial biofilm formation, and maintaining the structural integrity of mature biofilm. LapA is presumed to be an essential adhesin for attachment and biofilm formation, while LapF is characterised as an adhesin for cell–cell interaction in mature biofilm. However, the role of LapA in biofilm appears to be more critical [1,2,3,4].

The transport and exposure of *Pseudomonas fluorescens* LapA onto the cell surface have been extensively studied. Due to the similarity of the system to *P. putida*, it can be considered as a model for both bacteria. The LapA of *P. fluorescens* is transported to the cell surface via a type I secretion system (T1SS) that consists of three proteins, LapB, LapC and LapE [5]. LapB is expected to recognise the C-terminal signal sequence of LapA. The C-terminus of the protein is transported out of the cell, while the N-terminus remains in the periplasm [6]. LapA remains tethered to the outer membrane via LapE, inducing attachment to biotic and abiotic surfaces [7,8,9,10]. The periplasmic protease LapG releases LapA from LapE after the proteolytic attack of the N-terminus of LapA, thereby reducing the biofilm. Depending on the presence of c-di-GMP, LapD, submerged to the inner membrane, can inhibit the LapG activity by binding to it [11]. Unlike *P. fluorescens*, the *P. putida* LapA transporter genes are not located consecutively; *lapBC* (PP_0167 and PP_0166) form an operon, whereas *lapE* (PP_4519) is located separately. It is important to note that *lapA* and *lapB* of *P. putida* share the same, approximately 1 kbp long, promoter region, with the genes encoded in the opposite direction [12].

The LapAs are large extracellular proteins; in *P. fluorescens* Pf0-1 and *P. putida* KT2440 the size of the protein is 519 kDa and 888 kDa, respectively [12]. Based on the similarity of protein sequence, several domains have been found in LapAs that could be involved in its transport and adhesion. While LapA N-terminus remains in the periplasm, the extracellular repeats make up the majority of LapA. Although these repeats are not conserved among LapA homologs, it is suggested that the functionally similar repetitions provide a tubular structure, ensuring the location of the C-terminus far away from the cell [13]. The C-terminus of LapA, the extracellular part of this protein, contains the T1SS signal sequence required to transport with the LapBCE complex and a repeat-in-toxin (RTX) domain, described as a domain that binds calcium [10,12,14]. One particular domain of the C-terminus, the von Willebrand factor A-like domain (vWFa), has our attention. In *P. putida* strain KT2440, amino acids 8159-8283 of LapA are predicted to form the vWFa-domain based on the Pfam database and *P. fluorescens* Pf0-1 in 4710-4830, respectively [12]. The vWFa of *P. fluorescens* LapA is associated with adhesion into hydrophilic surfaces [10], but so far, the function of *P. putida* vWFa has not been described. 

The von Willebrand factor (vWF) is a eucaryotic multimeric glycoprotein found in blood plasma, platelets, endothelial cells, and subendothelial cells. Its primary function is to induce the adhesion of platelets and thrombus formation in the case of vascular injury [15]. Several eukaryotic proteins, such as type IV collagen, type VII collagen, cartilage matrix protein, undulin, leukocyte β2 class receptors, complement components C2, B, etc., contain a domain A of vWF (vWFa). These proteins are mostly cell surface proteins or extracellular proteins [16]. In addition to eucaryotic proteins, domain A of the von Willebrand factor is found in several bacterial proteins, especially among extracellular adhesins such as LapA homologs [11]. Hence, domain A is particularly characteristic of extracellular proteins involved in cell adhesion, migration, homing, pattern formation, and signal transduction [16].

Chronic infection of all pathogenic bacteria begins with their attachment to the surface. Notably, bacteria have been reported to bind to vWF or have surface proteins with vWFa-like domains used for attachment. In any case, the protein–protein interaction plays a significant role in developing virulence. For example, *Staphylococcus aureus* uses Spa protein to attach to vWF or its proteolytic products. *S. aureus* can attach to several blood and matrix proteins; however, the vWF is a target in the early stage of infection [17]. It has been demonstrated that the vWFa-like domain plays a central role in triggering the virulence of pathogenic bacteria. For example, the activation of *Pseudomonas aeruginosa* virulence genes depends on two conditions: quorum sensing and the mechanical signal involved in recognising host cells. PilY1 is a *P. aeruginosa* surface protein with a vWFa domain. The deletion of this domain was accompanied by a loss of virulence of *P. aeruginosa*, and it was thought that the binding of the PilY1 vWFa domain to host cells was a mechanical signal that, in addition to the quorum-sensing signal, triggers the expression of virulence genes [18].

We have previously shown that the biofilm of *P. putida* depends on LapA and the proteinaceous component in the medium [1,19]. Specifically, according to the crystal violet absorption, *P. putida* forms more biofilm when tryptone or poly-lysine are present in the medium [19]. In addition, we have ascertained that Fis activates *lapA* transcription in *P. putida* [20]. We presumed that additional expression of *fis* from the P*_tac_* promoter by IPTG, i.e., an artificial increase in the amount of Fis in the cell (so-called *fis*-overexpression), has a positive epistatic effect on the *P. putida* biofilm (Fis-enhanced biofilm). However, we recently ascertained that the positive impact of Fis on the *P. putida* biofilm depends on the presence of peptides in the medium [19]. In other words, the overexpression of *fis* in cells can enhance biofilm by activating *lapA* transcription only when peptides are present in the medium [19].

In this study, we hypothesised that the extracellular peptide promotes *P. putida* biofilm via LapA. To test this, we constructed a strain of *P. putida* in which *lapA* transcription is under the control of the P*_tac_* promoter, and the addition of IPTG regulates its transcription. In other words, we subtracted the Fis-enhancing effect on the *P. putida* biofilm from the LapA effect. In addition, we deleted the vWFa-domain sequence from *P. putida lapA* in order to ascertain the involvement of the vWFa domain in peptide-dependency of biofilm and found that *P. putida* biofilm is amplified by extracellular peptides if the LapA contains the correct vWFa-domain.

## 2. Results

### 2.1. LapA Is Essential for Peptide-Dependent Biofilm

Knowing the peptide-dependency of *P. putida* biofilm [19], we tested whether it is based on LapA or LapF. We previously reported that LapA is a crucial factor in forming *P. putida* biofilm in LB and that LapF is required for *P. putida* cell surface hydrophobicity [1,2,20]. Thus, we assessed the biofilm formation of PSm and F15 deletant strains ΔlapA, ΔlapF, and double deletant ΔlapAF in the following media: M9-0.2CAA; M9-0.2CAA+tryptone, and LB (Appendix A and Figure 1).

The two terms are used hereafter: Fis-enhanced biofilm and peptide-dependent biofilm. Fis-enhanced biofilm appears in *P. putida* F15 strain when the expression of *fis* is increased by IPTG compared to the biofilm of F15 without IPTG supplementation (compare biofilms of *P. putida* F15 in Figure 1B,C). Peptide-dependent biofilm appears in *P. putida* strains when the peptides are present in growth media compared to the biofilms formed in growth media without peptides (compare biofilms in Figure 1A,C) [19]. If the arithmetic means of the samples are significantly different (*p* < 0.05), it is considered a Fis-enhanced or peptide-dependent biofilm. Otherwise, the increased amount of Fis in the cells or the peptides in the growth medium does not affect biofilm.

The F15 has been a helpful strain in evaluating the positive effects of peptides on *P. putida* biofilm. The Fis-enhanced biofilm was formed in the presence of the peptides [19], the *lapA* transcription was activated by *fis*-overexpression [20], and *lapF* transcription was repressed [21]. We hypothesised that if LapA or LapF are involved in the peptide-dependence of biofilm, then the peptide-dependence of biofilm should be absent in strains where *lapA* or *lapF* genes are deleted, even in cells with an elevated amount of Fis.

We observed that the deletion of *lapA* decreased the biofilm of *P. putida* wild-type strain PSm, regardless of the medium (Figure 1). The *lapA* deletion strain PSmΔlapA had 1.3–1.9 times decreased biofilm than the wild-type strain PSm (*p* < 0.032), except for the *lapA*, *lapF* double deletion derivative PSmΔlapAF in M9-0.2CAA+tryptone, whose biofilm was similar to PSm (Figure 1). The *lapF*-deletion strain PSmΔlapF formed biofilms similarly to the wild-type strain PSm in the presence of peptides in media (Figure 1B,C).

In the case of *fis*-overexpression strain F15, the Fis-enhanced biofilm appeared in the tryptone-containing growth media with a 2.3-fold difference in M9-0.2CAA+tryptone (*p* < 0.0001) and a 2.0-fold difference in LB (*p* < 0.0001), but not in M9-0.2CAA (1.1-fold difference, *p* = 1; Figure 1). The strain with the deleted *lapA* gene F15ΔlapA did not produce a Fis-enhanced biofilm (Figure 1B,C). At the same time, the *lapF* deletion strain F15ΔlapF retained the positive effect of *fis*-overexpression on the F15 biofilm in M9-0.2CAA+tryptone and LB (3.9 and 2.0-fold differences, respectively, *p* < 0.0001), indicating that *lapA* expression is needed for the enhanced biofilm by Fis in the presence of peptides in the medium (Figure 1B,C).

We previously observed the positive effect of poly-L-lysine (pK) on *P. putida* F15 biofilm; it restored the Fis-enhancing effect for *P. putida* strain F15 grown in the M9-0.2CAA medium [19]. We supplemented M9-0.2CAA with 0.4 mg/mL of pK and assessed the biofilm formation of *lapA* and *lapF* deletants of *P. putida* (Figure 2). The Fis-enhanced biofilm appeared only in the presence of functional LapA (Figure 2). LapF did not impact the Fis-enhanced biofilm (Figure 2). However, the Fis-enhancing effect was observed for the F15ΔlapAF double mutant (1.5-fold difference, *p* < 0.0012), probably due to an unknown mechanism rather than polypeptides or adhesins (Figure 2).

The results from *P. putida* wild-type strain PSm were more complex. Specifically, the presence of pK in M9-0.2CAA increased the biofilm of PSmΔlapA but not the biofilm of PSmΔlapF, which had functional LapA and lacked LapF (Figure 2). However, as the positive effect of pK appeared on PSmΔlapAF double mutant, similarly PSmΔlapA, it indicates that the impact of poly-lysine could be more complex, and besides the adhesins, more factors could influence the biofilm. 

### 2.2. The Transcription of lapA Is Growth Medium Independent, the Amount of LapA Is Not

Because the absence of LapA abolished the peptide-dependence effect on the *P. putida* biofilm (Figure 1), the question arose whether the peptide dependency on the *P. putida* biofilm could be caused via *lapA* expression. Accordingly, the *lapA* expression in *P. putida* was assessed at transcriptional and protein levels. The *lapA* transcription was evaluated by β-galactosidase assay, and the abundance of LapA protein was estimated in crude cell lysates (Figure 3). LacZ activity was measured using plasmid p9_PlapA1-8, which carries the *lapA* promoter region in front of the reporter gene *lacZ* (Table 1).

Our results revealed that transcription of *lapA* was dependent on the medium but did not correlate with the F15 biofilm (Figure 1 and Figure 3A) [19]. F15 revealed the positive effect of *fis*-overexpression only in the presence of peptides (Figure 1). In all media, LacZ activity in F15 in the presence of 1 mM IPTG was 1.26 (M9-0.2CAA), 1.3 (M9-0.2CAA+tryptone), and 1.48 (LB) times higher than in the non-IPTG variant (*p* < 0.0001), indicating that regardless of the growth medium, the *lapA* transcription was always increased in the presence of the higher amount of Fis (Figure 3A). LacZ activity in PSm or F15 containing the empty vector p9TTBlacZ was less than 0.5 Miller units.

The crude cell lysates were prepared from the bacteria grown for 18 h in the appropriate medium to quantify the adhesins LapA and LapF. Because cell lysates prepared from different media were electrophoresed on PAA gels separately, the ratio of adhesins with and without IPTG was compared separately in every medium. In contrast to the level of *lapA* transcription, the amount of LapA protein in the crude cell lysate was affected by the content of the growth medium of bacteria (Figure 3B–D).

Since the effect of Fis on the F15 biofilm was evident (Figure 1 and Figure 2), we focused on comparing the amount of LapA in F15. Overexpression of *fis* in the tryptone-containing medium increased the amount of LapA 1.50 times (M9-0.2CAA) and 1.46 times (LB) in the F15 strain, which correlated with the previously observed increase in F15 biofilm formation in the same growth conditions (Figure 1 and Figure 3C) [19]. However, the *fis*-overexpression did not elevate the amount of LapA in F15 grown in the M9-0.2CAA medium (Figure 3C). Thus, the unchanged amount of LapA in *fis*-overexpressing cells might be the reason why F15 did not form a Fis-enhanced biofilm in the tryptone-free M9 medium (Figure 1).

It is known that Fis represses *lapF* transcription [21]; therefore, we used LapF as a control for LapA measurements. Indeed, in contrast to LapA, the *fis*-overexpression decreased approximately 4 times the amount of LapF in F15 in every media compared, indicating no correlation with a media-dependent biofilm of *P. putida* F15 (Figure 1 and Figure 3D).

In conclusion, the reduced amount of LapA in crude cell lysate prepared from cells grown in a medium without tryptone correlates with a reduced amount of *P. putida* biofilm.

### 2.3. Construction of a Strain with a Modified lapA Transcription and a LapA Strain without a vWFa Domain

Although we knew that Fis increases *lapA* transcription and LapA is required for biofilm formation (Figure 1 and Figure 3), we had to split the Fis-effect from the LapA-effect on the biofilm. There are two reasons: (i) Fis can enhance biofilm through another factor than LapA, and (ii) *P. putida* forms the biofilm even in the absence of LapA in the cell [1,20]. Therefore, *P. putida* strains were constructed in which native *lapA* is under the P*_tac_* promoter, and IPTG controls its transcription. *P. putida* strain PaW85 (isogenic for KT2440), one of the standard strains of *P. putida*, was used for this purpose (Table 1). Additionally, the PSm derivative of the PaW85 was used to evaluate the biofilm in the context of PSm and F15 (Table 1).

The construction of the *lapA* expression system was problematic for several reasons. Firstly, *lapA* is a very long gene (26 kbp) [12]; therefore, we could not use vector systems. Secondly, chromosomal *lapA* shares a promoter region with its transporter genes *lapBC* encoded opposite to *lapA* [12,20]. The disruption of the joint promoter region can affect the transcription of both genes. We used several expression systems to control the transcription of both *lapA* and *lapBC* genes. For example, an expression system was constructed that ensured the expression of *lapA* from the P*_tac_* promoter by IPTG and the transcription of *lapBC* from the Pm promoter by benzoate (data not shown). In this case, transcription in both directions was regulated by extracellularly added effectors. Unfortunately, the addition of phenolic compounds to the medium affected the biofilm formation per se; these strains were excluded from the study (data not shown). We had to use a construct that had the P*_tac_* promoter immediately upstream of the lapA gene. The reversely expressed *lapBC* retained the entire native promoter area without the possibility of external regulation by effectors. With this system, we constructed a P-PANB strain from PaW85 (Table 1). This variant is not ideal because, despite induced *lapA* transcription, the effect on the biofilm may remain modest, as the natively expressed transporter may begin to limit LapA exposure on the outer membrane.

The third part of the LapA transporter, LapE, is described in *P. fluorescens* as an outer membrane subunit where LapA N-terminus will trap. In this way, the extracellular LapA is anchored in the outer membrane [32]. The *lapE* deletion prevents the LapA transport to the outside of the cell, and the biofilm should weaken. Thus, this deletant strain was used as a negative control for P-PANB strains. The *lapG* deletion, on the other hand, was used to construct a positive control. LapG is a periplasmic protease that, when activated, cleaves the N-terminus of LapA and releases LapA into the environment [32], resulting in a reduction of biofilm. Thus, the deletion of *lapG* should produce a robust *P. putida* biofilm, as LapA is not removed from the outer membrane. Before this research, we had constructed *lapE* and *lapG* deletion variants from *P. putida* strain PSm, PSm-E, and PSm-G, respectively. Therefore, we created the IPTG-inducible *lapA* transcription strains PSm-E-PANB and PSm-G-PANB, based on PSm-E and PSm-G, respectively (Table 1).

The sequence coding a von Willebrand factor type A (vWFa) domain in *P. putida lapA* was deleted (from amino acid S8167 to 8281P, designated Avwf^−^). Recent studies have shown that the deletion of the vWFa domain in *P. fluorescens* LapA reduces biofilm formation to glass and hydrophilic plastic [10,32]. The von Willebrand factor itself, after which the bacterial adhesin domains were named, is a multimeric glycoprotein important for the protein–protein interaction that causes thrombus formation and bleeding arrest after injury of eukaryotic tissue [15]. Inspired by these works, we asked whether the vWFa domain of *P. putida* LapA may have a role in enhancing peptide-mediated biofilm. In other words, is the LapA vWFa domain required for peptide-dependent biofilm formation.

First, it was necessary to evaluate the dependence of *lapA* expression on IPTG in the constructed strains. For this purpose, the amount of LapA from the crude cell lysates was assessed by SDS-PAA gel-electrophoresis. In all strains containing the P*_tac_* promoter in front of the *lapA* gene, LapA was visibly increased in all media by 1 mM of IPTG ( Figure 4A, Figure 5A, and Figure 6A). In addition to LapA, the unknown protein complexes were visible in these gels that could not be LapA. The unknown protein complexes (probably the complexes of ribosomes) migrate slower than LapA, appear irregularly, and appear in PSmΔlapA, PSmΔlapF (Figure 4A, Figure 5A, Figure 6A, Figure 7B, and Appendix A), and PSmΔlapAF strains (Appendix A). However, we cannot exclude that *lapA* would or would not be expressed in P-PANB without IPTG, as unknown protein complexes migrated higher or at approximately the same height as LapA in SDS-PAA gels ( Figure 4A, Figure 5A, and Figure 6A).

### 2.4. LapA Domain vWFa Is Involved in the Peptide-Dependent Biofilm of P. putida PaW85 

We were interested in the behaviour of *P. putida* PaW85-based strains. These strains do not contain mini-Tn7 insertion after the *glmS* gene, and *lapA* transcription is directly regulated by IPTG without indirect effects on the *fis*-overexpression in the cell (Table 1). It is essential to consider two facts: (i) LapA exposure on the outer membrane is complexly regulated and depends on proteolysis and the presence of effectors, (ii) the expression of LapA transporters (*lapBC* and *lapE*) is not regulated in these strains.

The biofilm was evaluated by every medium separately. In the M9-0.2CAA medium, the biofilm of all PaW85-based strains was similar (Figure 4B). Although P-PANB forms more biofilm in the presence of IPTG than in its absence (*p* = 0.025), P-PANB biofilm does not differ from P-PANB-Avwf^−^ biofilm (*p* > 0.108); there is no significant biological difference among the biofilms of these strains.

The importance of LapA became evident in the M9-0.2CAA+tryptone medium. Although the positive effect of *lapA* expression by IPTG on the P-PANB biofilm was modest (1.2-fold, *p* = 0.0004), the biofilm of bacteria without the LapA vWFa domain (P-PANB-Avwf^−^) was significantly decreased (Figure 4C). Strain P-PANB-Avwf^−^ had 1.4 times less biofilm in medium without IPTG than P-PANB (*p* < 0.0001) and 1.6 times less biofilm in medium with IPTG (*p* < 0.0001). Whereas the P-PANB biofilm was IPTG-dependent, the vWFa deletion biofilm was insensitive to *lapA* expression by IPTG (P-PANB-Avwf; Figure 4C). 

In the LB medium, the P-PANB biofilm was increased 1.3-fold in the presence of IPTG (*p* < 0.0001), whereas deletion of the vWFa domain resulted in the independence of IPTG (Figure 4D). The biofilm of the P-PANB-Avwf^−^ strain was similar regardless of the presence or absence of IPTG (Figure 4D). The biofilm of the original strain PaW85 did not depend on the vWFa domain (Figure 4D).

In a polylysine-containing M9-0.2CAA medium, IPTG did not affect the biofilm of the P-PANB strain (Figure 4E). Still, the biofilm of vWFa domain deletion strains (Avwf^−^) was decreased compared to the original strains from which they were constructed (Figure 4E). Independent of IPTG, the strain P-PANB-Avwf^−^ had 1.5 times less biofilm than the P-PANB strain (*p* < 0.0001, Figure 4E). For PaW85 versus P-Avwf^−^, deletion of the vWFa domain reduced biofilm 1.2-fold (*p* = 0.063) and 1.4-fold (*p* < 0.001) in the absence and presence of IPTG, respectively (Figure 4E).

Our previous study showed that adding water-soluble cellulose to a medium enhances *P. putida* biofilm [19]. Deleting the vWFa domain did not affect the biofilm in the M9-0.2CAA+cellulose medium; both vWFa-deletion strains behaved similarly to its origin strain, P-Avwf^−^ similarly to PaW85 and P-PANB-Avwf^−^ similarly to P-PANB (Figure 4F). However, the addition of IPTG to the medium increased 1.2-times biofilms of P-PANB strains compared to cells grown without IPTG, P-PANB (*p* = 0.006) and the P-PANB-Avwf^−^ strain (*p* < 0.0001; Figure 4F). Biofilms in the M9-0.2CAA+PGA medium were comparable and did not depend on the *lapA* expression with IPTG or the presence of the vWFa domain (Figure 4G).

### 2.5. Biofilm of P. putida PSmΔlapE- and PSmΔlapG-Derived Strains

We constructed a *lapE* deletion mutant from PSm, the PSm-E, which lacked the third subunit of LapA transporter, LapE (Table 1). The *lapE* deletion reduced 1.6 and 1.7-times the biofilm of PSm in the M9-0.2CAA medium with 1 mM IPTG (*p* < 0.0001) and without IPTG (*p* < 0.001) accordingly (Figure 5B).

Similar results were obtained in the M9-0.2CAA+tryptone medium, where the biofilm of PSm-E was decreased 1.6 and 1.7 times with and without the IPTG in the medium, respectively, compared to PSm (Figure 5C). However, in the LB medium *lapE* deletion did not affect the biofilm of *P. putida* wild-type strain PSm (Figure 5D).

Since the transporter of LapA was incomplete in Δ*lapE*-based strains, we expected that manipulating the *lapA* expression by IPTG or deleting vWFa would not affect the biofilm in these strains. Indeed, the IPTG did not affect PSm-E-PANB and PSm-E-PANB-Avwf^−^ biofilm in any of the used media (Figure 5B–D). However, in the M9-0.2CAA medium, for some reason that we cannot explain, the vWFa deletion reduced 1.3 times PSm-E-PANB biofilm (*p* < 0.001; Figure 5B).

Deletion of *lapG* should trap all LapA transported on the outer membrane and enhance the biofilm. Indeed, all strains with *lapG* deletion (Figure 6) have a more robust biofilm than the wild-type PSm in all media (Figure 5). The presence of 1 mM IPTG in the growth medium increased PSm-G-PANB biofilm compared to the absence of IPTG increased biofilm 2.27 times in M9-0.2CAA (*p* < 0.0001), 1.6 times in M9-0.2CAA+tryptone (*p* < 0.0001) and 1.23 times in LB (*p* < 0.0001; Figure 6B–D). The deletion of the vWFa domain from LapA (P-PANB- Avwf^−^) did not reduce the biofilm to the same level as that detected in the PSm-Avwf^−^, indicating that LapA in significant excess may affect biofilm without the vWFa domain (Figure 5 and Figure 6). Additionally, the PSm-G-PANB strain without IPTG had a relatively robust biofilm compared to PSm in the presence of tryptone in media (Figure 5C,D and Figure 6C,D). These results suggest the P*_tac_* promoter may still leak. However, we cannot state that this would result in a relatively robust biofilm of PSm-G-PANB in media without IPTG.

### 2.6. The vWFa Domain Is Essential for F15 Biofilm

We hypothesised that the Fis-enhanced biofilm of *P. putida* F15 in LB medium is caused, at least in part, by the positive effect of Fis on *lapA* expression [1,20]. Although the effect of Fis on *lapA* transcription was modest, the *lapA* deletion in F15 vanished this effect, indicating that LapA could be involved in the biofilm enhancement by Fis (Figure 1) [1]. Therefore, we constructed an F15-Avwf^−^ strain in which the *fis* was possible to overexpress by IPTG, and LapA lacked the vWFa-domain. We expected to see the reduced effect of *fis*-overexpression on the biofilm in this strain.

The *fis*-overexpression in F15 and F15-Avwf^−^ strains by IPTG was assessed in M9-0.2CAA, M9-0.2CAA+tryptone and LB media (Figure 7A). The expression of *fis* was similar in both strains in all media tested, and IPTG induced the expression of *fis* from the P*_tac_* promoter in these strains (Figure 7A). After that, the LapA presence in the crude cell lysates of F15-Avwf^−^ strains was assessed (Figure 7B). The ratio of LapA in cells grown in the presence of 1 mM IPTG versus without IPTG was similar to F15; 1.21 (95% confidence interval of arithmetical mean 0.60) in M9-0.2CAA, 2.04 (1.48) in M9-0.2CAA+tryptone and 1.58 (1.22) in LB medium. Although the differences in LapA ratios of F15-Avwf^−^ were statistically insignificant in the media assessed, LapA ratios increased similarly to F15 when IPTG and tryptone were added to the media (Figure 3C).

F15 and F15-Avwf^−^ strains formed biofilm similarly in M9-0.2CAA, without a Fis-enhanced biofilm (Figure 7C). Despite the similar amount of LapA and Fis in both strains, differently from the strain F15, the IPTG did not improve F15-Avwf^−^ biofilm in M9-0.2CAA+tryptone or LB medium F15 (Figure 7C). In contrast, the deletion of the vWFa domain did not abolish the Fis-enhanced biofilm in the M9-0.2CAA+poly-L-lysine medium (Figure 7C). This could be explained by a strong positive charge effect of poly-L-lysine that binds negatively charged LapA, regardless of the vWFa-domain.

## 3. Discussion

The *P. putida* biofilm is a dynamic phenotype depending on the growth phase of bacteria and the environment where the bacterium lives [1,19]. Our previous research showed that peptides in growth media increase the *P. putida* biofilm [19]. In addition to the fact that the peptides may act as an additional source of C and N, the peptides may induce a more substantial biofilm by being a structural component of the biofilm matrix [19].

To study biofilm, we used *P. putida* strain F15, which carries the additional *fis* gene under the control of P*_tac_* (IPTG). We used the *fis*-overexpression strain F15 because the use of strains lacking or down-regulating *fis* was not possible due to the essentiality of the *fis* gene or instability of the constructed strain. Surprisingly, this artificial strain served as an excellent tool for identifying the factors influencing *P. putida* biofilm. Although it is an overexpression strain, it is possible to study the physiology of the bacterium. For example, Fis represses transcription of *lapF* through a single binding site in front of *lapF*, thereby reducing cell surface hydrophobicity [2,21]. Mutation of the Fis-binding site restored *lapF* transcription, amount of LapF in cells, and ultimately, surface hydrophobicity [2,21]. Hence, this direct effect of Fis on the bacterial phenotype through one binding site in front of another gene encouraged us to use F15 in LapA expression studies as well.

The main aim of this study was to assess the involvement of surface adhesins, LapA and LapF, in the peptide-dependent *P. putida* biofilm. We knew that the peptides increase the biofilm of the *P. putida* wild-type strain PSm, and the so-called Fis-enhancing effect on the F15 biofilm only occurs in the presence of the peptides [19]. Moreover, we assessed the Fis-enhancing effect on biofilm in LB and suggested that the searchable factor could be LapA, as Fis activates *lapA* transcription and increases the amount of LapA in F15 in LB [1,20]. Surprisingly, Fis activates *lapA* transcription in all studied growth media, but the increased quantity of LapA and Fis-enhanced biofilm appeared when growth media were supplemented with peptides (Figure 1, Figure 2 and Figure 3). It indicates that in the absence of peptides in growth media, the post-transcriptional regulatory steps could affect the expression of *lapA* more intensively. For example, in the absence of peptides in the growth medium, the translation of *lapA* may be downregulated, LapA itself may be effectively degraded, or LapA may be released from the surface due to the activity of LapG. We can only speculate which factor might cause the peptide-dependent post-transcriptional regulation of *lapA* expression. For example, the RsmA, a component of the GacS/A signalling pathway, binds RNAs and can regulate translation initiation, RNA stability, and/or transcript elongation [33] and the involvement in the regulation of the GacS/A signalling pathway *P. putida lap*-genes expression has been previously shown [34,35]. One of three RsmA homologs in *P. putida*, RsmE, down-regulates *lapA* expression in LB medium [35]. On the other hand, LapA can be efficiently released from the outer membrane due to the increased amount of active LapG in the periplasm. The amount of active LapG depends not only on the expression of the *lapGD* operon, the amount of protease in the periplasm, but also on the effectors such as c-di-GMP and ppGpp; and the availability of calcium [4,36,37]. However, we cannot exclude the possibility that other regulators are involved in the peptide-dependent expression of *lapA*. Thus, the post-transcriptional steps in *lapA* expression, including translation and the amount of LapA on the cell surface, may depend on the components of the growth medium.

The importance of LapG proteolytical activity appears in comparing P-PANB, PSm-E-PANB, and PSm-G-PANB biofilms in the growth medium without tryptone (M9-0.2CAA). Namely, LapA quantity is elevated in all strains by IPTG supplementation (Figure 4A, Figure 5A, and Figure 6A), while only PSm-G-PANB, the *lapG* deletion strain, had a robust biofilm in the M9-0.2CAA medium (Figure 4B, Figure 5B and Figure 6B). Thus, proteolytic cleavage of LapA from the outer membrane probably affects peptide-dependent biofilm formation more than the transporters. However, we cannot exclude the possibility that P-PANB biofilm is a result of a lack of LapA transporters, indicating peptide dependent regulation of *lapBC* and/or *lapE* genes.

The P-PANB strain constructed from PaW85 had only a modest, albeit significant, effect on biofilm in peptide-containing media upon induction of *lapA* transcription by IPTG (Figure 4). However, the Avwf^−^-strains of PaW85, without the vWFa domain in LapA, had a significantly decreased biofilm relative to the strains with the correct LapA (Figure 4). The importance of the vWFa-domain to the biofilm was particularly evident when P-PANB and P-PANB-Avwf^−^ were compared. Both strains had *lapA* transcription under the control of P*_tac_* promoter (IPTG). Still, the absence of the vWFa-domain in LapA eliminated the effect of the elevated amount of LapA by IPTG in the media containing the peptides (Figure 4). We decided to use previously constructed PSmΔlapE and PSmΔlapG strains as a negative and positive control of inducible *lapA* expression. In the first case, increasing the amount of LapA or the absence of the vWFa domain should not significantly affect the biofilm because in the absence of LapE, LapA does not reach the cell surface. Indeed, in general, the biofilm of *lapE*-minus strains did not depend on the induction of *lapA* transcription by IPTG or the absence of vWFa (Figure 5). In addition, experiments on the *lapG* deletion strain confirm the importance of the vWFa domain for *P. putida* biofilm formation (Figure 6). The LapA strain without the vWFa domain resulted in less biofilm than the strains with correct LapA, although the absence of *lapG* always produced a stronger biofilm than the wild-type strain (Figure 5 and Figure 6). In sum, we can conclude that not only does the presence of LapA cause the peptide-dependent *P. putida* biofilm, but LapA must also have the correct vWFa domain.

Since the absence of the LapA vWFa domain negatively affected the biofilm in *P. putida* wild-type strains, we investigated the effect of the lack of this LapA domain on the F15 biofilm (Figure 7). So far, we hypothesised that, among other factors, Fis could increase the F15 biofilm via *lapA* because Fis activates *lapA* transcription to some extent [20]. The complete loss of the Fis-enhancing effect for biofilm in LB and M9-0.2CAA+tryptone medium was astonishing when we deleted the vWFa-domain sequence from the F15 *lapA* gene, F15-Avwf^−^ (Figure 7). Therefore, the enhancing effect of the Fis to *P. putida* biofilm described so far must be entirely or predominantly dependent on LapA. Consequently, Fis enhances *lapA* transcription, at least to some extent, and the LapA vWFa domain increases biofilm when peptides are present in the medium.

However, the question arises why the lack of vWFa domain had a pronounced impact on the F15 biofilm, whereas wild-type PaW85 with elevated LapA had only a modest effect (Figure 4 and Figure 7). It seems that Fis may more broadly control the *lapA* expression and its persistence on the cell surface. For example, Fis may indirectly regulate LapA persistence on the cell surface through other genes, whereas the wild-type strain undergoes native strong down-regulation. As a result, the wild-type bacterium may lose most of the expressed *lapA* to the environment. Indeed, according to our preliminary data, Fis represses transcription of the *lapGD* operon (data not shown), and it may explain at least partly a substantial biofilm enhancement with elevated Fis in *P. putida* F15. However, the transport and maintenance of LapA on the cell surface seems to be more critical for biofilm than transcriptional regulation. Consequently, the role of Fis in regulating the expression of other *lap* genes or the peptide-dependent regulation needs to be further investigated.

The LapA vWFa domain of *P. fluorescens* has been previously described as essential for hydrophilic surface binding [10,32]. Our research has shown the importance of poly-L-lysine, a hydrophilic positively charged biomolecule for *P. putida’s* LapA vWFa-domain (Figure 7). Notably, hydrophilic Na-cellulose and PGA, which are negatively charged biopolymers, have no direct relationship to biofilm enhancement via vWFa (Figure 4F,G). Referring to our previous study, neither uncharged poly-DL-alanine nor negatively charged poly-L-glutamate had a biofilm-enhancing effect on *P. putida* F15 [19]. However, the positively charged spermine enhanced *P. putida* F15 biofilm similarly to poly-L-lysine [19].

Moreover, the F15-Avwf^−^ strain entirely or largely lost the positive effect of the peptides on biofilm formation (Figure 7C). Therefore, the surface’s hydrophilicity and positive charge appear to be necessary for the binding LapA by the vWFa domain to the surface. However, we cannot say whether vWFa is specific for a particular sequence or whether it binds to a surface only by its electrostatic forces, which is an issue for future research to explore.

However, how precisely LapA binds to the surface has not been elucidated. One explanation may be the importance of other LapA domains for surface binding. For example, the F15ΔlapA and F15-Avwf^−^ biofilms in the medium M9-0.2CAA+pK form biofilm differently (Figure 2 and Figure 7C). The complete absence of LapA eliminates the Fis-enhancing effect for biofilm, whereas LapA without the vWFa domain is significantly reduced but does not entirely abolish it. Thus, binding through another LapA domain may be necessary. However, it may simply be an electrostatic effect appearing in excess of adhesin and biopolymer. Due to the increased amount of Fis in *P. putida* F15, the cell surface remains hydrophilic, the amount of LapA increases, and the amount of LapF decreases drastically (Figure 3) [2]. The resulting biofilm formation may be promoted through non-specific binding between the positively charged poly-lysine and LapA.

In conclusion, this research was initiated because we had previously determined that LapA, extracellular peptides, and increased Fis in the cell promoted *P. putida* biofilm formation. The question was whether these factors are interrelated. The LapA domain of vWFa was found to be essential for biofilm enhancement when peptides were added to the medium. It was particularly evident in *P. putida* with an increased amount of Fis. This so-called biofilm-enhancing Fis effect wholly or partially depends on the LapA vWFa domain. In addition, it seems that the most crucial regulatory step in the *lapA* expression and maintenance of LapA on the cell surface is the post-transcriptional and not the transcriptional step.

## 4. Materials and Methods

### 4.1. Bacterial Strains, Plasmids, and Media

The bacterial strains and plasmids used in this study are described in Table 1 and used oligonucleotides in Appendix A. Bacteria were grown in a complete LB medium containing 10 g/L tryptone (LabM, Heywood, UK), 5 g/L yeast extract (LabM, Heywood, UK) and 5 g/L NaCl, or in media based on M9 buffer [38]. The M9-0.2CAA medium contained M9 buffer [38], 2.5 mL/L of microelements [39], 2 g/L of glucose and 2 g/L of CAA (casamino acid; Difco, Detroit, MI, USA) and 0.01 g/L tryptophan. M9-0.2CAA was amended with 0.4 mg/mL poly-L-lysine (1000–5000 Da; Sigma-Aldrich, St. Louis, MO, USA); or biopolymers: with 10 g/L tryptone (LabM, Heywood, UK), with 0.4 g/L the sodium salt of carboxyl methylcellulose (Sigma-Aldrich, St. Louis, MO, USA) or PGA (poly-galacturonic acid). All used growth media are summarised in Appendix A.

*P. putida* was incubated at 30 °C, *E. coli* at 37 °C. Solid media contained 1.5% agar (Difco, Detroit, MI, USA). Antibiotics were added at the following concentrations: 10 µg gentamicin mL^−1^, 100 µg ampicillin mL^−1^, 50 µg kanamycin mL^−1^, 1.5 mg benzylpenicillin mL^−1^, 200 µg streptomycin mL^−1^.

### 4.2. DNA Manipulation

Vectors for PANB and Avwf^−^ construction were created by circular polymerase extension cloning (CPEC) protocol [40]. The vector pEMG-lapB-*lapA* was made from the pEMG backbone, omitting the *lacZ* DNA [28] (Table 1). Therefore, the pEMG and two fragments, the 5′ terminus of *lapA* and *lapB*, were amplified together. The pEMG was amplified with oligonucleotides 1.2-EMG5′ and 3.1-lapB3′-SceI-EMG3′ (Appendix A). The oligonucleotides 5.1-lapB5-DS-SD-lapA5′ and 2.3.-EMG-SceI-lapA3′ were used to amplify the first 743 nucleotides of *lapA* 5′ terminus from *P. putida* PaW85 chromosome (Appendix A). The oligonucleotides 4.1-lapB3′ and 6.1-lapB5′ were used to amplify the first 750 nucleotides of *lapB* 5′ terminus from *P. putida* PaW85 chromosome (Appendix A). The resulting vector pEMG-lapB-lapA carried oppositely orientated *lapA* and *lapB* 5′ terminuses (Appendix A).

The vector pEMG-lapB-Pm-xylS-lacI-P*_tac_*-lapA was amplified together from pEMG-lapB-lapA and two fragments, one carrying the lacI-P*_tac_* cassette and the other holding the xylS-Pm cassette (Table 1). The oligonucleotides 11-Pm-DS-lapB5′ and 12-P*_tac_*-SD-lapA5′ were used for pEMG-lapB-lapA amplification, 9-T1-xylS-lacI3′ and 10-lapA5′-DS-P*_tac_* for *lacI*-P*_tac_* cassette amplification from the vector pGP-FpFy-Km-lacItac-lapFSD, and 7-lapB5′-DS-Pm and 8-lacI3′-xylS-T1 for xylS-Pm cassette amplification from the vector pSEVA228S [21,29] (Appendix A). The resulting vector pEMG-lapB-Pm-xylS-lacI-P*_tac_*-lapA carried the P*_tac_* promoter in front of the 5′ terminus of the *lapA* gene and the Pm promoter in front of the 5′ terminus of *lapB* (Appendix A).

The final vector pEMG-P*_tac_*A-natB used to construct the PANB-strains was generated from the backbone of pEMG-lapB-Pm-xylS-lacI-P*_tac_*-lapA, omitting the xylS-Pm cassette (Table 1). The oligonucleotides 1.3.-(lap)Aprom3′-T1 and 3.3.-(lap)Aprom5′-EMG3 were used for amplification of pEMG-lapB-Pm-xylS-lacI-P*_tac_*-lapA backbone, and 13.-(lap)Aprom3′ and 14.-EMG3-(lap)Aprom5′ were used for amplification of *lapAB* promoter-area from *P. putida* PaW85 chromosome, 770 nucleotides from *lapA* gene. The resulting vector pEMG-P*_tac_*A-natB carried P*_tac_* promoter in front of *lapA* 5′ terminus, *lacI* gene, and 770 nucleotides of *lapAB* promoter-area (Appendix A).

The vector pGNW-lapA-Avwf^−^, for deletion of potential vWFa domain encoding DNA in *lapA* gene, was made from the pGNW2 backbone, omitting the *lacZ* DNA [30] (Table 1 and Appendix A). Therefore, the pGNW2 and two fragments, the 5′ and 3′ DNA from the vWFa domain of *lapA*, were amplified together. The oligonucleotides 3.5-A5vwf5-SceI-EMG3 and 1.2-EMG5′ were used for pGNW2 amplification (Appendix A). The 883-bp-long 5′ DNA from the potential vWFa domain was amplified with the oligonucleotides 4.5.-A5vwf5 and 6.5.1.-A3vwf5-A5vwf3 from the *P. putida* PaW85 chromosome (Appendix A). The 806-bp-long 3′ DNA from the vWFa domain was amplified with 5.5.1.-A5vwf3-A3vwf5 and 2.5.-EMG-SceI-A3vwf3 from the *P. putida* PaW85 chromosome (Appendix A).

For the construction of *lapG* (PP_0164) and *lapE* (PP_4519) deletion mutants, the DNA regions that flanked the deleted genes were cloned into the suicide vector pEMG [28]. *P. putida* PaW85 chromosome was used for the 470 bp long upstream DNA and 607 bp long downstream DNA of *lapG* amplification with oligonucleotides PP0164-I-fw and PP0164-I-rev, and PP0164-2-fw and PP0164-2-rev, respectively. These fragments were ligated together by overlap extension-PCR and cloned into pEMG using BamHI and EcoRI endonucleases, resulting in pEMG-ΔlapG [41] (Table 1). The suicide vector pEMG-ΔlapE was constructed similarly to pEMG-ΔlapG (Table 1). The 503 bp long upstream DNA and 680 bp long downstream DNA were amplified with oligonucleotides PP4519-I-fw-EcoRI and PP4519-I-rev, and PP4519-2-fw and PP4519-2-rev-BamHI, respectively, ligated together and cloned to pEMG using BamHI and EcoRI endonucleases, resulting in pEMG-ΔlapE (Appendix A).

The deletion strains and *lapA* expression strains were constructed according to a previously published protocol [28]. To this end, constructed pEMG vectors (Table 1) were electroporated [42] into the corresponding *P. putida* strain to obtain a cointegrate. Plasmid pSW was then introduced into the cells, which generated deletion variants from the strains by activating homing endonuclease SceI expression [27]. All vectors and modifications on the *P. putida* chromosome were checked by sequencing to exclude PCR and recombination errors (Appendix A).

### 4.3. Assessment of Biofilm

Biofilm formation was monitored in flat-bottom microtiter plates (Greiner Bio-One GmbH, Kremsmünter, Austria; 96-well cell culture plate Cat-No 655180) by the Fletcher method [43]. A 100 μL measure of LB or M9-0.2CAA medium with or without biopolymers and 1 mM of IPTG was placed in the well and inoculated with 5 μL overnight-grown bacteria. Bacteria were grown for 24 h at 30 °C and stained with 25 μL of 1% crystal violet for 15 min. The wells were washed twice with 200 μL of water, and crystal violet was extracted twice with 180 μL of 96% ethanol. The absorption (A_540_) of three times diluted crystal violet extracts in water was measured by a microtiter plate reader Tecan Sunrise-basic [25]. Antibiotics were added to the growth medium only for the pre-growing of inoculum; a biofilm assay was performed without antibiotics. At least three independent measurements were performed.

### 4.4. Assessment of β-Galactosidase Activity

An indirect method, β-galactosidase activity, was used to assess *lapA* transcription in different media in the presence and absence of 1 mM IPTG for 18 h at 30 °C. LacZ activity was measured in *P. putida* PSm and F15 containing the p9_P*lapA*1-8 or p9TTBlacZ plasmids (Table 1); therefore, the growth media was supplemented with 1.5 mg benzylpenicillin mL^−1^ [20]. The β-galactosidase measurements in cell suspension were performed according to Sambrook et al. (1989) [44]. At least three independent measurements were performed.

### 4.5. Preparation of Cell Lysates, Protein Quantification, SDS-Polyacrylamide Gel Electrophoresis, and Western Immunoblot Assay

*P. putida* strains were grown in 5 mL of the respective medium without antibiotics for 18 h in the presence and absence of 1 mM IPTG [1]. For the preparation of crude cell lysate, the cells were collected by centrifugation and suspended in RIPA buffer (50 mM Tris/HCl pH 7.4, 1% Triton X-100, 0.1% SDS, 0.5% sodium deoxycholate, 1% NP-40, 500 mM NaCl [45]), 100-fold diluted Halt protease and phosphatase single-use inhibitor cocktail (Thermo Fisher Scientific, Waltham, MA, USA) and 5 mM EDTA, and incubated for 4 h at 37 °C [1]. The cells’ debris was separated by centrifugation at 12,000× *g* for 15 min at 4 °C [1]. The total amount of protein in the cleared supernatant was measured spectrophotometrically by the content of tryptophan [46]. Then, 40 μg and 10 μg of total protein were diluted 1:1 in 2× Laemmli sample buffer for protein quantification and Western immunoblot assay, accordingly, and incubated for 30 min at 37 °C [1]. For quantification, the proteins were separated on a 4–8% gradient PAA gel at 4 °C overnight by electrophoresis at 20 V using the following running buffer: 37.5 mM Tris, 95 mM glycine, and 0.05% SDS. After that, proteins in SDS-PAA gels were silver strained [47] and the intensity of LapA and LapF bands were quantified by ImageQuantTM TL 8.2 software.

For Western blotting, 10 µg of total proteins and 45 ng of *P. putida* Fis(His_6_) as a positive control [48] were loaded per well, separated by 12% Tricine-SDS-PAGE [49], and transferred to a PVDF membrane (Amersham Hybond-P; Cytiva. Marlborough, MA, USA). The membrane was probed with monoclonal mouse anti-Fis antibodies for Western blotting at a final dilution of 1:2000, followed by alkaline phosphatase-linked goat anti-mouse IgG dilution of 1:10,000 (LabAs Ltd., Tartu, Estonia) and developed using 5-bromo-4-cloro-indolyl phosphate/nitro blue tetrazolium.

### 4.6. Statistical Analysis of the Results

Factorial ANOVA was used to assess the variability of data in experiments. The multiple comparisons of means were conducted following Bonferroni’s test for unequal n. For statistical tests, the significance level was set at *p* < 0.05. Homogenous groups with the same letter indicate similarity of the arithmetic means (*p* ≥ 0.05). The calculations were performed using Statistica 13 software.

## Figures and Tables

**Figure 1 ijms-23-05898-f001:**
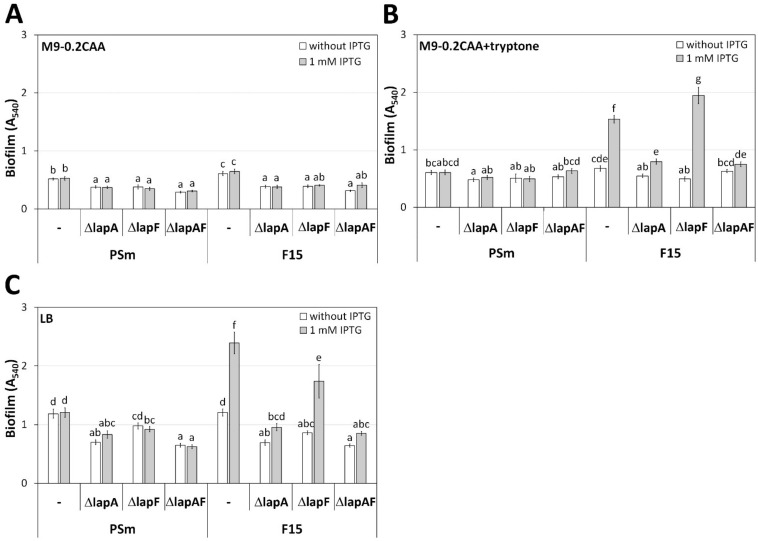
Biofilm of *P. putida* strains PSm, F15 and deletion-mutants ΔlapA, ΔlapF, and ΔlapAF constructed from them. “-“ indicates original strain without deletions. Cells were grown in (**A**) M9-0.2CAA, (**B**) M9-0.2CAA+tryptone, and (**C**) LB medium. Cells were grown with or without 1 mM IPTG. Arithmetic means of at least three independent sets of measurements are shown. The 95% confidence intervals of the arithmetic means and the homogeneous groups are shown above the columns by lower-case letters, tested by following Bonferroni’s test for unequal n. Identical letters in homogenous groups denote nonsignificant differences (*p* ≥ 0.05) between averages of biofilm.

**Figure 2 ijms-23-05898-f002:**
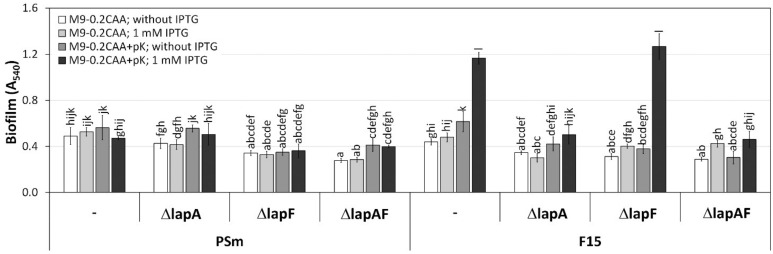
Biofilm of *P. putida* strains PSm, F15, and deletion-mutants ΔlapA, ΔlapF, and ΔlapAF constructed from them in the M9-0.2CAA medium with poly-L-lysine (pK) or without it. “-“ indicates original strain without deletions. Cells were grown with or without 1 mM IPTG. Arithmetic means of at least three independent sets of measurements are shown. The 95% confidence intervals of the arithmetic mean and the homogeneous groups are shown above the columns by lower-case letters, tested by following Bonferroni’s test for unequal n. Identical letters in homogenous groups denote nonsignificant differences (*p* ≥ 0.05) between averages of biofilm.

**Figure 3 ijms-23-05898-f003:**
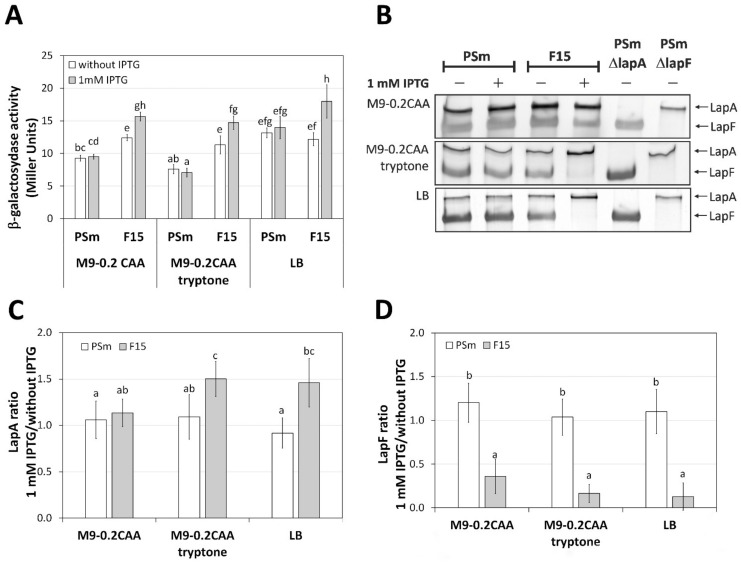
The expression of adhesins LapA and LapF in *P. putida* strains PSm and F15. (**A**) β-galactosidase activity in cells carrying pB_P*lapA*1-8 where *lapA* promoter region cloned in front of *lacZ* gene. Arithmetical means of at least three independent sets of measurements and 95% confidence intervals of the arithmetic mean. The homogenous groups are shown above columns by lower-case letters. Identical letters denote nonsignificant differences (*p* ≥ 0.05) among averages of means. (**B**) SDS-PAA gel electrophoresis from cell lysates prepared from cells grown in M9-0.2CAA, M9-0.2CAA+tryptone or LB medium with or without IPTG. PSmΔlapA and PSmΔlapF cell lysates were loaded onto the gel as negative controls of LapA and LapF. The arrows indicate the location of LapA and LapF. 40 mg of total protein was applied to the line. (**C**) Normalised LapA and (**D**) LapF amount. The amount of adhesine determined from the crude lysate of cells grown with IPTG was divided by the amount of LapA determined from the crude lysate of cells grown without IPTG. The averages of at least 6 biological parallels are given, with 95% confidence intervals. The homogenous groups are shown above columns by lower-case letters. Identical letters denote nonsignificant differences (*p* ≥ 0.05) between the averages of the ratios.

**Figure 4 ijms-23-05898-f004:**
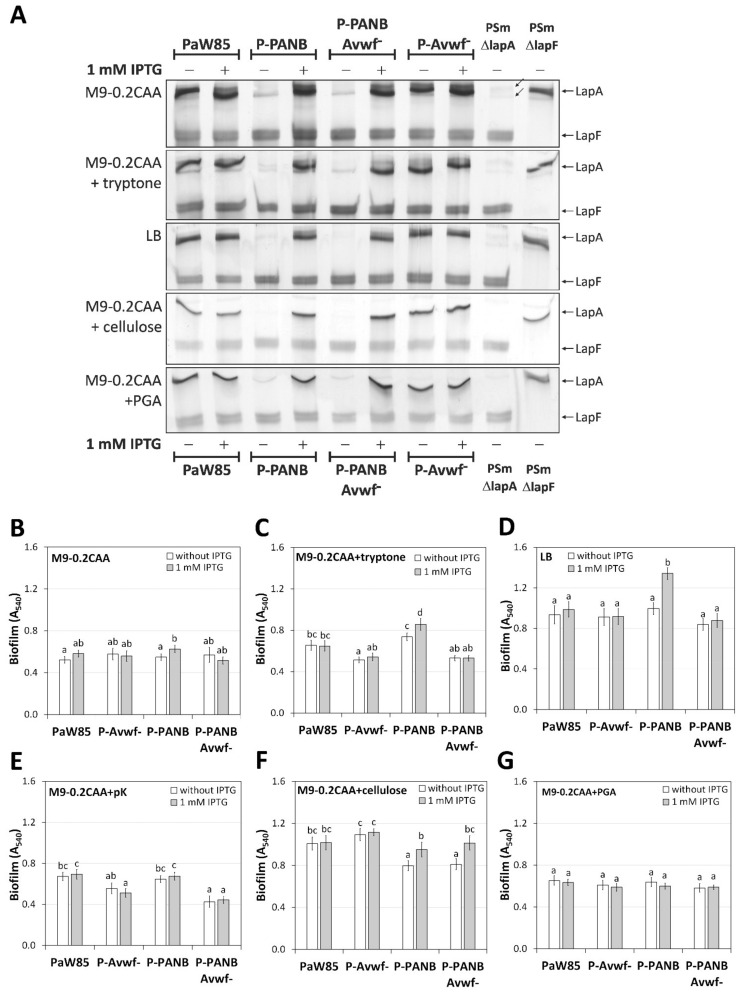
LapA and LapF amount in *P. putida* PaW85-based strains and biofilm. (**A**) SDS-PAA gel electrophoresis of crude cell lysates, 40 μg of total protein is loaded to each lane. The media in which cells were grown for preparing crude cell lysates are shown on the left of the gel image. Cells were grown with and without 1 mM IPTG. The strains used in the analysis are shown above and below the figure. PSmΔlapA and PSmΔlapF strains were used as negative controls for LapA and LapF. Arrows indicate LapA and LapF on the right side of the figure. Inclined arrows indicate an unknown protein complex. (**B**) Biofilm of PaW85-based strains in M9-0.2CAA, (**C**) M9-0.2CAA+tryptone, (**D**) LB, (**E**) M9-0.2CAA+poly-L-lysine, (**F**) M9-0.2CAA+cellulose and (**G**) M9-0.2CAA+PGA in the presence and absence of 1 mM IPTG. Arithmetical averages of at least three biological parallels with 95% confidence intervals are given. Homogeneity groups are shown above the columns by lower-case letters. Identical letters denote nonsignificant differences (*p* ≥ 0.05) between averages of biofilm. Multifactorial ANOVA was used for statistical analysis, analysed separately by media.

**Figure 5 ijms-23-05898-f005:**
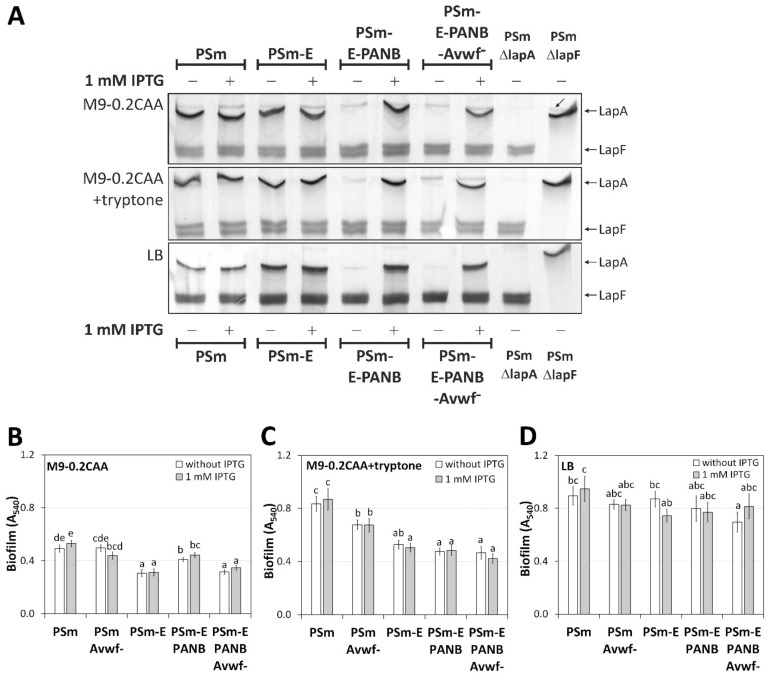
SDS-PAA gels and biofilm of *P. putida* PSm and PSm-E based strains and biofilm. (**A**) SDS-PAA gel electrophoresis of crude cell lysates, 40 μg of total protein is loaded to each lane. The media in which cells were grown for preparing crude cell lysates are shown on the left of the gel image. Cells were grown with and without 1 mM IPTG. The strains used in the analysis are shown above and below the figure. PSmΔlapA and PSmΔlapF strains were used as negative controls for LapA and LapF. Arrows indicate LapA and LapF on the right side of the figure. Inclined arrows indicate an unknown protein complex. (**B**) Biofilm of PSm-based strains in M9-0.2CAA, (**C**) M9-0.2CAA+tryptone and (**D**) LB in the presence and absence of 1 mM IPTG. An arithmetical average of at least three biological parallels with 95% confidence intervals are given. Homogeneity groups are shown above the columns by lower-case letters. Identical letters denote nonsignificant differences (*p* ≥ 0.05) between averages of biofilm. Multifactorial ANOVA was used for statistical analysis, analysed separately by media.

**Figure 6 ijms-23-05898-f006:**
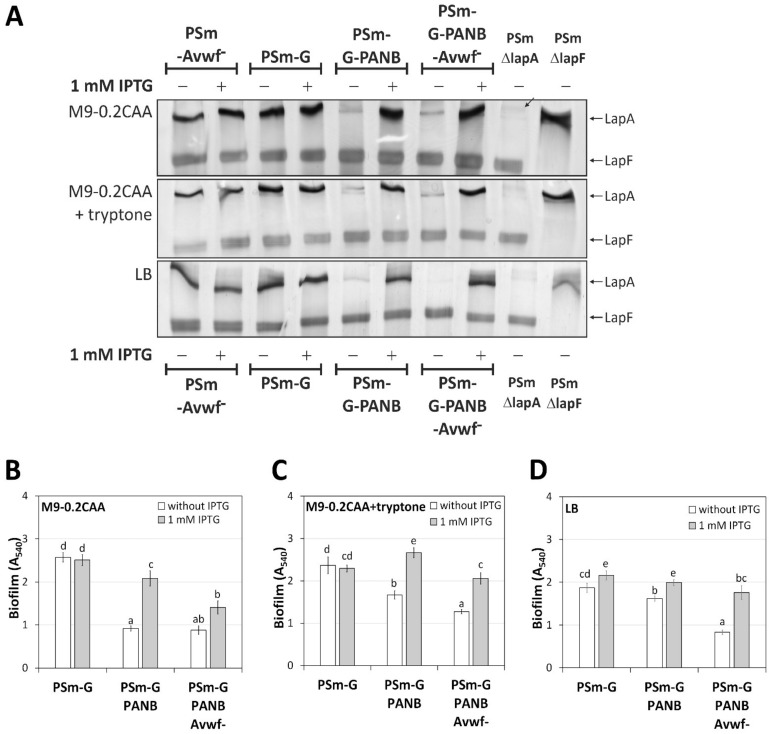
SDS-PAA gels and biofilm of *P. putida* PSm-G-based strains and biofilm. (**A**) SDS-PAA gel electrophoresis of crude cell lysates, 40 μg of total protein is loaded to each lane. The media in which cells were grown for preparing crude cell lysates are shown on the left of the gel image. Cells were grown with and without 1 mM IPTG. The strains used in the analysis are shown above and below the figure. PSmΔlapA and PSmΔlapF strains were used as negative controls for LapA and LapF. Arrows indicate LapA and LapF on the right side of the figure. The inclined arrow indicates an unknown protein complex. (**B**) Biofilm of PSm-G-based strains in M9-0.2CAA, (**C**) M9-0.2CAA+tryptone and (**D**) LB in the presence and absence of 1 mM IPTG. An arithmetical average of at least three biological parallels with 95% confidence intervals are given. Homogeneity groups are shown above the columns by lower-case letters. Identical letters denote nonsignificant differences (*p* ≥ 0.05) between averages of biofilm. Multifactorial ANOVA was used for statistical analysis, analysed separately by media.

**Figure 7 ijms-23-05898-f007:**
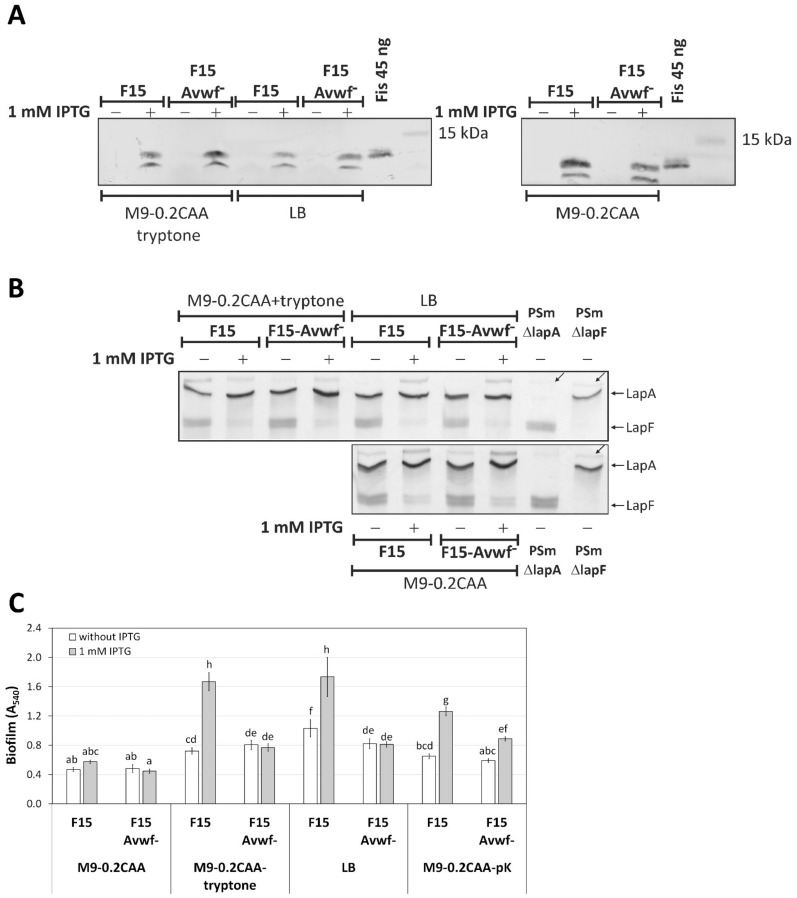
Expression of *fis* in F15-based strains by IPTG, the presence of LapA and LapF in cells and biofilm. (**A**) Expression of *fis* in *P. putida* F15 by immunoblotting using monoclonal anti-Fis antibodies. 10 μg of total protein obtained from *P. putida* F15 grown in LB, M9-0.2CAA or M9-0.2CAA+tryptone media was loaded on a 12% SDS-PA gel. The supplementation of 1 mM IPTG is shown by “+” above the lane. 45 ng of purified Fis (6His) was used as a positive control. Protein marker band sizes are indicated. (**B**) SDS-PAA gel electrophoresis of crude cell lysates, 40 μg of total protein is loaded to each lane. The media in which cells were grown for preparing crude cell lysates are shown to the left of the gel image. Cells were grown with and without 1 mM IPTG. The strains used in the analysis are shown above and below the figure. PSmΔlapA and PSmΔlapF strains were used as negative controls for LapA and LapF. LapA and LapF are indicated by arrows on the right side of the figure. Inclined arrows indicate an unknown protein complex. (**C**) Biofilm of *P. putida* F15-based strains. An arithmetical average of at least three biological parallels with 95% confidence intervals are given. Homogeneity groups are shown above the columns by lower-case letters. Identical letters denote nonsignificant differences (*p* ≥ 0.05) between averages of biofilm. Multifactorial ANOVA was used for statistical analysis, analysed separately by media.

**Table 1 ijms-23-05898-t001:** Bacterial strains and plasmids used in this study.

Strain or Plasmid	Genotype or Description	Reference
** *E. coli* **		
CC118 λpir	Δ(*ara-leu*) *araD* Δ*lacX74 galE galK phoA20 thi-1 rpsE rpoB argE* (Am) *recA1* *λ**pir* phage lysogen	[22]
** *P. putida* **		
PaW85	isogenic to KT2440	[23,24]
P-PANB	PaW85; term-*lacI*^q^-*P_tac_*-*lapA*	This study
P-PANB-Avwf^−^	PaW85; term-*lacI*^q^-*P_tac_*-*lapA*ΔS8167-8281P	This study
P-Avwf^−^	PaW85; *lapA*ΔS8167-8281P	This study
PSm	PaW85, isogenic to KT2440; chromosomal mini-Tn7-ΩSm1 (Sm^r^)	[25]
PSmΔlapA	PSm; ΔPP_0168 (Sm^r^)	[1]
PSmΔlapF	PSm; ΔPP_0806 (Sm^r^)	[1]
PSmΔlapAF	PSm; ΔPP_0168 ΔPP_0806 (Sm^r^)	[1]
PSmΔlapE	PSm; ΔPP_4519	This study
PSmΔlapG	PSm; ΔPP_0164 (Sm^r^)	This study
PSm-Avwf^−^	PSm; *lapA*ΔS8167-8281P	Thus study
PSm-E-PANB	PSm; ΔPP_4519 (Sm^r^), term-*lacI*^q^-P*_tac_*-*lapA*	This study
PSm-E-PANB-Avwf-	PSm; ΔPP_4519 (Sm^r^), term-*lacI*^q^-P*_tac_*-*lapA*ΔS8167-8281P	This study
PSm-G-PANB	PSm; ΔPP_0164 (Sm^r^), term-*lacI*^q^-P*_tac_*-*lapA*	This study
PSm-G-PANB-Avwf^−^	PSm; ΔPP_0164 (Sm^r^), term-*lacI*^q^ P*_tac_*-*lapA*ΔS8167-8281P	This study
F15	PaW85, isogenic to KT2440; chromosomal mini-Tn7-ΩGm-term-*lacI*^q^-P*_tac_*-*fis*-T1T2 (Gm^r^)	[25]
F15ΔlapA	F15; ΔPP_0168 (Gm^r^)	[1]
F15ΔlapF	F15; ΔPP_0806 (Gm^r^)	[1]
F15ΔlapAF	F15; ΔPP_0168, ΔPP_0806 (Gm^r^)	[1]
F15-Avwf^−^	F15; *lapA*ΔS8167-8281P	This study
**Plasmids**		
p9_PlapA1-8	951 bp long promoter-area of *lapA* cloned in front of *lacZ* (Amp^r^)	[20]
p9TTBlacZ	A promoter probe vector containing *lacZ* reporter gene (Amp^r^); RK2-based vector	[26]
pSW	I-SceI-expressing plasmid (Amp^r^)	[27]
pGP-FpFy-Km-lacItac-lapFSD	R6K suicide vector pGP704-L containing P*_tac_* promoter with *lacI* repressor gene (Amp^r^, Km^r^)	[21]
pEMG	R6K suicide vector (Km^r^)	[28]
pEMG-lapB-lapA	Derivate of pEMG containing in the opposite direction of 5′ ends of *lapA* and *lapB* genes (Km^r^)	This study
pEMG-lapB-Pm-xylS-lacI-P*_tac_*-lapA	Derivate of pEMG-lapB-*lapA* containing Pm promoter in front of *lapB* 5′ end and P*_tac_* promoter in front of *lapA* 5′ end, *xylS* and *lacI* genes (Km^r^)	This study
pEMG-P*_tac_*A-natB	Derivate of pEMG-lapB-*lapA* containing P*_tac_* promoter in front of *lapA* gene and native promoter region in front of *lapB* gene, *lacI* gene (Km^r^)	This study
pSEVA228S	oriV(RK2), *xylS*, Pm→I-SceI; Km^r^	[29]
pGNW2	Derivative of pEMG carrying P14g→msfGFP (Km^r^)	[30]
pGNW-lapA-Avwf^−^	Derivate of pGNW2 containing *lapA* DNA for deletion of vWFa domain in *lapA* gene	This study
pUTmini-Tn5 Km2	Suicide vector, source of Km resistance gene (Amp^r^, Km^r^)	[31]
pEMG-ΔlapE	Derivate of pEMG containing *lapE* (PP_4519) flanking DNAs (Km^r^)	This study
pEMG-ΔlapG	Derivate of pEMG containing *lapG* (PP_0164) flanking DNAs (Km^r^)	This study

## Data Availability

All data generated or analysed during this study are included in this published article.

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
