# Peer review of "Pseudomonas putida Biofilm Depends on the vWFa-Domain of LapA in Peptides-Containing Growth Medium"

_ijms, 2022, doi:10.3390/ijms23115898_

Round 1

Reviewer 1 Report

I have carefully read the submission entitled 'Pseudomonas putida biofilm depends on the vWFa-domain of LapA in peptides-containing growth medium' by Puhm and colleagues. It falls into the scope of IJMS. The manuscript is very interesting and well organized, but hard to follow and lack precision of statements.  

The methods need to be provided with more precise descriptions. The authors do not bother to give details, just state 'as described previously'.

Due to high number of strains/mutants  and media used the manuscript is hard to follow. It would be advisable to add a table (in paragraph 4.1 or as supplemental material) summarising media used and introducing their abbreviations used through out the whole manuscript. Similarity, the strains designations need to be written through out the whole manuscript as they are in Table 1. Eg., in lines 109-110 authors write that 'P. putida lapA and lapF knock-out deletants' were used without provision of their designations as in Table 1 and in figures designations used are sometimes split and omitting delta letter (Δ) introducing much confusion to the reader (e.g., Fig. 1). 

Were the antibiotics used at the stage of strain genetic modifications only or throughout the all experiments. In case of the latter scenario there were no controls for their influence on the key processes researched with no GM but for antibiotic resistance only.

The language style need to be changed to be more precise and scientifically sound. What the authors mean by: 'an enhanced biofilm was formed' (line 126, was it just of the same size as in control?), 'We supposed that' (line 128), 'the deletion of lapA harms the biofilm of P. putida wild type strain' (line 131 - the lapA deletion can abrogate/ abolish, decrease biofilm formation by...), 'a weaker biofilm' (line 132 - provide extent in numbers as measured experimentally), 'tended' (line 134 - sounds as the results obtained gave inconsistent data), 'enhanced biofilm' (lines 138, 139, ...), 'did not produce an enhanced biofilm' (line 141; should that mean that the authors were expecting different outcome, but the biofilm formed was of the same size/density as in case of...), etc.

In Results (2.2.) the authors evaluate Plap transcriptional activity indirectly from plasmid construct that cannot substitute the promoter potential on chromosomal location. In this case a RT-qPCR should be performed. How the expression levels were normalised?

There are also several minor points that need to be revised:

lines 97-98 - the authors state that they '...constructed a wild-type strain of P. putida in...' ?! this is GM strain. 'Wild-type' is, by definition, a native strain isolated from environment. 

Table 1 - provide reference to prove that PaW85 strain is isogenic to KT2440

The titles of paragraphs in the Results should describe findings not experiment performed.

The whole manuscript must be rewritten.

Author Response

Dear Reviewer,

We're glad you took the time to read our manuscript and make suggestions for improving it. The manuscript has been improved and here are our answers.

  • Methods are rewritten (chapters 4.3, 4.4 and 4.5)
  • All used growth media are corrected and summarised in Table S2. The names of strains are written uniformly in the whole manuscript (for example, PSmΔlapAF), and Figures 1 and 2 are corrected.

  • All biofilms and growth for preparation of crude cell lysates or Western blot assay the bacteria were grown without antibiotics. The antibiotics were used for only the pre-growth of bacteria to avoid contamination. The only exception was the LacZ assay when it was necessary to keep plasmid carrying promoter area and reporter gene fusion in cells.

    The section on methodologies has been updated; chapters 4.3, 4.4 and 4.5.

  • The manuscript is improved according to your instruction. Since we are writing about many factors and mechanisms at once in the manuscript, a definition of Fis-enhanced and peptide-dependent biofilm is included at the beginning of the Result chapter to facilitate reading. All arithmetic means and significant differences of means (written as homogenous groups) are presented in figures; the necessary biofilm differences with p-values are included in the text. We avoid using the term "biofilm formation" because it is a broad term and can be understood as biofilm development that we did not study. Only one timepoint, a 24-hour-old biofilm, has been assessed in this manuscript.

  • There is no plausible qPCR reference standard because Fis increases stable RNA expression (and several common standards used for qPCR) in bacteria (for example [1]). Therefore, the accuracy of the qPCR results obtained from F15 grown with and without IPTG would be questionable.

    We have assessed the ratio of rRNA/cell dry mass and found that fis-overexpression can increase rRNA amount in P. putida cells up to 2 times. We have used several reference genes for qPCR, for example, rpoD, and monitored the expression of fis, lapA, rpoH, lapF (and expression of reference genes themselves), and the results did not overlap. Therefore, we cannot use this method.

    We have previously measured lapA and lapF transcription by LacZ-assay and ascertained that if we mutated the Fis-binding sites in the promoter area, the differences in LacZ activities equalised, indicating loss of Fis regulation [2,3]. Fis binding to mutated DNA was also absent or reduced by the Dnase I footprint assay [2,3]. It indicates that LacZ-assay is a suitable alternative method for assessing the Fis effect on transcription in a fis-overexpression background. The fact that the regulation of lapA expression may be more complex in M9-0.2CAA medium than in LB is fascinating. As a result, we plan to investigate this in the future. 

    The LacZ measurements from cell suspension were performed according to the protocol of Sambrook et al. (1989) [4] using the following equation:

    LacZ activity = (1000 × (A420 – 1.75 × A550)) / (t × V × A600)

    A420 is the accumulation of ortho-nitrophenol, A550 cell debris in the reaction mixture, t time of reaction in minutes, V volume of bacterium culture in the reaction mixture and separately measured A600 bacterial culture absorption used for LacZ-assay.

    The reference is corrected in the manuscript.

  • References are provided.

  •  The titles of paragraphs in the Result chapter are changed.

  •  The manuscript is improved according to your instruction.
  1. Richins, R.; Chen, W. Effects of FIS overexpression on cell growth, rRNA synthesis, and ribosome content in Escherichia coli. Biotechnology progress 2001, 17, 252-257, doi:10.1021/bp000170f.
  2. Ainelo, H.; Lahesaare, A.; Teppo, A.; Kivisaar, M.; Teras, R. The promoter region of lapA and its transcriptional regulation by Fis in Pseudomonas putida. PloS one 2017, 12, e0185482, doi:10.1371/journal.pone.0185482.
  3. Lahesaare, A.; Moor, H.; Kivisaar, M.; Teras, R. Pseudomonas putida Fis binds to the lapF promoter in vitro and represses the expression of LapF. PloS one 2014, 9, e115901, doi:10.1371/journal.pone.0115901.
  4. Sambrook, J., Fritsch, E.F. and Maniatis, T. . Molecular Cloning: A Laboratory Manual, 2nd ed.; Published by Cold Spring Harbor Laboratory Press: New York, USA: 1989.

Reviewer 2 Report

In this study Puhm et al., describes the role of LapA and molecular mechanisms involved in the media induced biofilm formation in Pseudomonas putida. Overall this is an interesting study and suitable for the publication in IJMS. However, there are some concerns with the manuscript that the authors need to address before considering for publication.

1) Page 5 Results section 2.2: The authors have used a reporter assay for evaluating the transcription of LapA, which is not ideal, particularly when they compare it with protein expression levels. The authors need to perform quantitative PCR assays to determine the amount of LapA mRNA expressed.

In contrary to the reporter assay there is a significant difference in protein levels are evident under various media conditions. The authors have used whole cell lysate for the protein quantitation. It is plausible that lapA mRNA is translated into protein but the enhanced expression of periplasmic protease LapG secrete LapA into the media and consequently inhibit biofilm formation. Therefore, the authors need to compare the amount of LapA secreted into the media for better understanding of the observed phenomenon. Further, it would be ideal if the authors could provide the LapG expression under these conditions.  

2) There are several vague statements given without much supporting data and based on pure speculations such as “The presence of tryp- 225 tone in the medium affects the post-transcriptional expression stages of lapA”. The authors need caution while giving such vague statements and are recommended to remove or rewrite such statements.  Besides, there are several instances the authors have used “PSmΔlapAlapF” for double deletion mutants. The authors are suggested to rectify this typo, I assume the authors intended to use “Δ” instead of “A”.

Author Response

Dear Reviewer,

We're glad you took the time to read our manuscript and make suggestions for improving it. The manuscript has been improved, and here are our answers.

1) There is no plausible qPCR reference standard because Fis increases stable RNA expression (and several common standards used for qPCR) in bacteria [1]. Therefore, the accuracy of the qPCR results obtained from F15 grown with and without IPTG would be questionable.

We have assessed the ratio of rRNA/cell dry mass and found that fis-overexpression can increase rRNA amount in P. putida cells up to 2 times. We have used several reference genes for qPCR, for example, rpoD, and monitored the expression of fis, lapA, rpoH, lapF (and expression of reference genes themselves), and the results did not overlap. Therefore, we cannot use this method

We have previously measured lapA and lapF transcription by LacZ-assay and ascertained that if we mutated the Fis-binding sites in the promoter area, the differences in LacZ activities equalised, indicating loss of Fis regulation [2,3]. Fis binding to mutated DNA was also absent or reduced by the Dnase I footprint assay [2,3]. It indicates that LacZ-assay is a suitable alternative method for assessing the Fis effect on transcription in a fis-overexpression background. The fact that the regulation of lapA expression may be more complex in M9-0.2CAA medium than in LB is fascinating. As a result, we plan to investigate this in the future.

- Indeed, the regulation of lap-genes expression is complex and needs to be understood to put the puzzle of P. putida biofilm together. However, to maintain the compactness, we focused on the involvement of adhesins and LapA vWFa domain in peptide-dependent biofilm. Moreover, we did not use the strains with enhanced lapG expression; the lapG-deletion strain PSmDlapG and its derivates were used. In the strains, PSm-G-PANB and PSm-G-PANB-Avwf- were lapG deleted, and these strains contained Ptac promoter in the front of lapA gene, not lapG. As we mentioned in the Discussion section, we left the study of post-transcriptional regulation of lapA to the future.

Yes, this topic is exciting, and, as usual, it raises many additional questions and hypotheses. For example, is the release of LapA regulated differently in planktic and sessile cells or how released LapA behave in the biofilm matrix? Will be it used as a structural component for LapAs in the outer membrane, etc.? The identification of lapGD operon's promoters in P. putida, ascertainment of expression regulation by extracellular factors and transcriptional regulators, and the release of LapA from the outer membrane in response to environmental signals are our current studies, and we plan to publish data in future.

2) The speculative interpretation of results is removed or changed to conditional in the Results chapter.

The names of strains are written uniformly in the whole manuscript (for example, PSmΔlapAF).

  1. Richins, R.; Chen, W. Effects of FIS overexpression on cell growth, rRNA synthesis, and ribosome content in Escherichia coli. Biotechnology progress 2001, 17, 252-257, doi:10.1021/bp000170f.
  2. Ainelo, H.; Lahesaare, A.; Teppo, A.; Kivisaar, M.; Teras, R. The promoter region of lapA and its transcriptional regulation by Fis in Pseudomonas putida. PloS one 2017, 12, e0185482, doi:10.1371/journal.pone.0185482.

3.            Lahesaare, A.; Moor, H.; Kivisaar, M.; Teras, R. Pseudomonas putida Fis binds to the lapF promoter in vitro and represses the expression of LapF. PloS one 2014, 9, e115901, doi:10.1371/journal.pone.0115901

Round 2

Reviewer 1 Report

Authors introduced all necessary changes in the text, corrected figures and tables. 

line 258 - consider removing 'could' as the LapA correlate with biofilm. Whether the correlation is due to analysed factors or by other (unrelated) mechanism the authors can elaborate in Discussion.

Author Response

The „could“ is removed. We have discussed these possibilities in the Discussion chapter (lines 484-508). 

Reviewer 2 Report

The authors have tried to address my concerns. However, I still have some concerns with the the data given on the the Fig 3 A, as it indicates that there is an enhanced transcription of the LapA (based on the reporter assay) even in media devoid of peptides in F15 strains with IPTG induction. Results from their previous study and data given on Fig 1 clearly indicate that this transcription does not lead to media induced Biofilm formation. While it may not be plausible for the authors to quantify the amount of LapA transcribed using qPCR accurately, the authors need to provide some plausible explanation why this increased transcription doesn't lead to protein production and biofilm formation. The data given in the Fig 3 B and C clearly indicates that increased transcription of the Lap A leads to the overall increase in the LapA protein in all the other conditions. One potential possibility is that LapA protein levels are high even in M9-0. 2CAA media, as with other media conditions. However, it is possible that that enhanced production of LapG protease inhibits the biofilm production in M9-0.2CAA. Therefore, it is likely that key determinant in the media/peptide induced Biofilm formation is lapGD operon. While the details of how LapGD operon are regulated can be a part of another manuscript, it is essential for this manuscript to have the information on the secreted LapA under various media conditions, as the LapA on the outer membrane is key for the LapA mediated biofilm formation. 

Author Response

Because lapA transcription in strain F15 is medium-independent, but protein amount in the crude cell lysate is dependent, it can be due to three reasons: 1) decreased translation of lapA, which is probably regulated by the Rsm system, 2) increased degradation of protein or 3) altered persistence of LapA in the cell membrane. All regulatory stages are post-transcriptional and should be considered as there is no indication that the decreased amount of LapA in the cell lysate is due to the increased expression of lapG.

We asked if there was a correlation between LapA amount in crude cell lysate and biofilm; yes, it was. We do not ask why the LapA amount in crude cell lysate is decreased in M9-0.2CAA because it is a separate piece of research, and it cannot be answered by measuring the amount of LapA in the supernatant alone. Whether the reduced biofilm is due to the release of LapA into the environment (activation of LapG without expression change or regulation of lapGD expression), increased degradation of protein, or reduced translation is irrelevant, in point of view this manuscript, as all results would be in the same direction. Measurement of LapA in the supernatant would be another experiment in this manuscript that may or may not explain the results. The decreased LapA amount may not be caused by LapG but by another step of expression such as down-regulation of lapA translation. Thus, the reason for the reduced amount of LapA in cells grown in the M9-0.2CAA will not change the result that LapA and, more precisely, its vWFa domain are responsible for peptide-dependent biofilm of P. putida. And LapGD will be a regulatory factor for peptide-dependent biofilm as the F15-Avwf- (expresses lapA without vWFa domain) has lost Fis-enhanced biofilm in the presence of peptides.

Round 3

Reviewer 2 Report

I suggest the authors to include the possible explanation provided in the their previous authors response for the low LapA protein observed despite efficient transcription of the gene in the discussion part of the manuscript with suitable references.  

Author Response

The discussion section has been updated to highlight possible mechanisms that may reduce the amount of LapA in cells grown in the peptide-free growth medium. (Lines 484-516)